# Dual Graph Diffusion Model for Social Recommendation

## Abstract

Graph-based social recommender systems utilize user-item interaction graphs and user-user social graphs to model user preferences. However, their performance can be limited by redundant and noisy information in these two graphs. Although several recommender studies on data denoising exist, most either rely on heuristic assumptions, which limit their adaptability, or use a single model that combines denoising and recommendation, potentially imposing substantial demands on the model capacity. To address these issues, we propose a dual Graph Diffusion Social Recommender (GDSR), which consists of two steps: graph denoising and user preference prediction. *First*, we design a denoising module which exploits a dual diffusion model to alleviate noises in the interaction and social graphs by performing multi-step noise diffusion and removal. We develop three kinds of conditions to guide our dual graph diffusion paradigm and propose a cross-domain signal guidance mechanism to enhance the structure of denoised graphs. *Second*, we devise a recommender module that employs a dual graph learning structure on denoised graphs to generate recommendations. Moreover, we use additional supervision signals from the diffusion-enhanced data augmentation to introduce a graph contrastive learning task, enhancing the recommender module's representation quality and robustness. Experiment results show the effectiveness of our GDSR. We release the anonymous code for reproducibility at https://anonymous.4open.science/r/GDSR-www.

## Keywords

Recommender Systems, Denoising.

## 1 Introduction

Social recommender systems, which are designed based on social influence and homophily theories [30, 31], utilize user-item interaction data and user social networks for recommendation. Early efforts utilize matrix factorization to integrate social data into user-item interaction modeling [11, 16, 28]. Recently, graph-based social recommenders, which utilize graph neural networks (GNNs) to model user high-order preferences and social influence propagation in the interaction bipartite graph and social graph structures, have become mainstream and achieved impressive progress [21, 37].

Despite their effectiveness, the performance of graph-based social recommenders can be limited by the presence of redundant and noisy information in both the user-item interaction graph and the social graph. Specifically, interaction data may not accurately

reflect users' true preferences due to inadvertent or erroneous interaction behaviors [32, 40]. Furthermore, recommender models may be biased by fake interactions created by malicious users [10, 57]. Additionally, social relations might also be inaccurately established, as users on social media might unintentionally follow or be followed by bots or fake accounts [24, 29], introducing bias into user modeling. Compared to traditional social recommenders, graph-based ones are potentially more vulnerable to such noise due to their neighborhood aggregation-based message-passing mechanism in GNNs, which enlarges the impact of noise on learning user and item representations. Thus, enhancing the robustness of graph-based social recommenders against noise in graphs is crucial.

Previous methods to reduce noise in recommendation data are typically classified into two categories [58]. One school is based on data cleaning, which includes resampling [3, 4] and reweighting [34, 44, 47] strategies. Resampling identifies noise and focuses on cleaner data for model training, while reweighting uses all the data but assigns lower weights to potentially noisy data. However, since these methods often depend on heuristic assumptions [44, 47] related to noise distribution, data cleaning-based methods require fine-grained adjustments to suit different backend models or datasets, limiting their adaptability [58]. The second research line is from the model perspective, enhancing the inherent noise resistance of recommenders [9, 48, 51]. Specifically, these methods first augment data by adding random noise [51] or discarding positive signals [9, 48], and then train recommender models with the augmented data to learn robust data representations. However, these model perspective-based approaches rely on a single model to reduce noisy data and generate recommendations, which may impose substantial demands on the model representation ability.

Diffusion model (DM) is a powerful type of generative model that has achieved state-of-the-art results in various research domains [25, 35, 36, 41]. DMs operate through forward and reverse processes, both of which inherently enhance denoising capabilities [13, 58]. In the forward process, DMs continuously introduce noise with controllable scales, which increases noise diversity. In the reverse process, DMs simplify the denoising task by breaking it down into multiple steps, each reducing the denoising difficulty. Recently, several studies attempt to integrate DMs with recommender systems [20, 22, 45, 58]. However, we believe they are not well-suited for graph-based social recommendation. Specifically, they do not focus on denoising from a graph structure perspective, and they lack specific adaptations (e.g., condition guidance) designed for the social recommendation task. The above analysis inspired us to design a DM paradigm specifically tailored for social recommendation.

To this end, in this paper, we propose a new graph-based social recommendation method called dual Graph Diffusion Social Recommender (GDSR), which contains a denoising module and a recommendation module. (1) In the denoising module, we design a dual DM structure, including a collaborative diffusion model (CDM) and a social diffusion model (SDM), to denoise the user-item interaction graph and the user-user social graph. Specifically, CDM

and SDM first corrupt the initial interaction and social graphs by gradually injecting Gaussian noises, respectively. After multiple noise accumulation, CDM and SDM iteratively remove noises by using a denoising neural network to generate denoised interaction and social graphs. To guide the reverse denoising process, we design three types of conditional information: global graph semantics, personalized information, and cross-domain knowledge signals, rather than relying solely on pure noise. Additionally, we develop a signal guidance mechanism that establishes collaboration between the CDM and SDM, leveraging cross-domain semantic signals to improve the denoised graph structures. (2) In the recommendation module, based on the denoised graphs, we employ a dual GNN equipped with a gated interaction mechanism to learn user and item representations for recommendation. The gated interaction mechanism facilitates knowledge sharing during the learning processes of two graphs. Moreover, we introduce a diffusion-aware graph contrastive learning task that enhances the representation quality and robustness of the recommendation module based on diffusion-enhanced data augmentation. We conduct extensive experiments on three datasets, and the results show that our GDSR outperforms several strong baselines. Ablation studies and case analyses are further performed to better understand our GDSR design and demonstrate the effectiveness of its key modules.

To summarize, our contributions in this work are as follows:

- We propose a social recommender GDSR, which integrates denoising and recommendation, enhancing data and recommendation quality via a dual DM and graph learning architecture.
- We develop a graph denoising approach that leverages dual DMs with the tailored guidance strategy for social recommendation, effectively alleviating noise in interaction and social graphs.
- We introduce a dual GNN structure with a diffusion-aware graph contrastive learning paradigm to model user preferences.

## 2 Preliminary

In this section, we first present some key definitions, followed by an introduction to the background of the diffusion model.

### 2.1 Notation and Problem Formulation

This section introduces the key concepts of the paper and formulates the definition of the social recommendation task. We detail the key notation table of this paper in **Appendix A**.

**User-Item Interaction Graph**. Given a user set $\mathcal{U}$ and an item set $\mathcal{V}$, we define the interaction matrix $\mathbf{Y} \in \mathbb{R}^{|\mathcal{U}| \times |\mathcal{V}|}$, where an entry $y_{uv} = 1$ denotes an interaction relation between user $u \in \mathcal{U}$ and item $v \in \mathcal{V}$, and a value of zero indicates no interaction. From the perspective of graphs, we could transform interaction matrix $\mathbf{Y}$ into bipartite graph structure $\mathcal{G_B} = (\mathcal{U} \cup \mathcal{V}, \mathbf{Y})$.

**User-User Social Graph**. The user social relation can be represented as a matrix $\mathbf{S} \in \mathbb{R}^{|\mathcal{U}| \times |\mathcal{U}|}$, with $s_{uu'} = 1$ showing a follow or trust relation between users $u$ and $u'$, and zero otherwise. Similarly, matrix $\mathbf{S}$ can be converted into a social graph $\mathcal{G_S} = (\mathcal{U}, \mathbf{S})$.

**Task Description**. Given the user-item interaction graph $\mathcal{G_B}$ and user-user social graph $\mathcal{G_S}$, our goal is to first generate two denoised graphs $\mathcal{G_{B^*}}$ and $\mathcal{G_{S^*}}$, and then learn a prediction function $\mathcal{F}: \hat{y}_{uv} = \mathcal{F}(u, v | \Theta, \mathbf{Y}, \mathcal{G_{B^*}}, \mathcal{G_{S^*}})$, where $\hat{y}_{uv}$ is the probability that $u$ will engage with $v$, and $\Theta$ is the parameter of function $\mathcal{F}$.

## 2.2 Diffusion Model

Diffusion model (DM) includes forward and reverse processes.

• **Forward process.** Given a data sample $x_0$, the forward process gradually introduces Gaussian noise with controllable scales and steps over $T$ iterations, increasing noise diversity [13, 45, 58]. Specifically, the transition from $x_{t-1}$ to $x_t$ is defined as follows:

$$q(x_t | x_{t-1}) = \mathcal{N}(x_t; \sqrt{1 - \beta_t} x_{t-1}, \beta_t I), \tag{1}$$

where $t \in \{1, 2, \cdots T\}$ is the diffusion step, $\beta_t \in (0, 1)$ is the Gaussian noise scale introduced at each step $t$, $I$ is the identity matrix, and $\mathcal{N}$ is the Gaussian distribution and it is used to sample $x_t$.

• **Reverse process.** This process iteratively denoises the noisy data $x_T$ according to the sequence $x_T \to x_{T-1} \cdots \to x_0$ [13, 45, 58]. Specifically, DMs learn the denoising process $x_t \to x_{t-1}$, as follows:

$$p_\theta(x_{t-1} | x_t) = \mathcal{N}(x_{t-1}; \mu_\theta(x_t, t), \Sigma_\theta(x_t, t)), \tag{2}$$

where $\mu_\theta(x_t, t)$ and $\Sigma_\theta(x_t, t)$ are the mean and covariance values, predicted by a neural network with parameters $\theta$. To maintain training stability, the learning of $\Sigma_\theta$ is commonly ignored [13], while mean $\mu_\theta$ can be further factorized as follows:

$$\mu_\theta(x_t, t) = \frac{1}{\sqrt{\alpha_t}} \left( x_t - \frac{1 - \alpha_t}{\sqrt{1 - \bar{\alpha}_t}} \epsilon_\theta(x_t, t) \right), \tag{3}$$

where $\alpha_t = 1 - \beta_t$, $\bar{\alpha}_t = \prod_{t'=1}^{t} \alpha_{t'}$ and $\epsilon_\theta$ learns to predict the source noise $\epsilon \sim \mathcal{N}(0, I)$ determining $x_t$ from $x_0$ [26].

• **Training.** The denoising neural network $\theta$ can be trained using the following simplified objective function [14, 54]:

$$\mathcal{L}_\theta = \mathbb{E}_{t, \epsilon \sim \mathcal{N}(0, I)} \left[ \| \epsilon - \epsilon_\theta(x_t, t) \|_2^2 \right], \tag{4}$$

where $t$ is sampled from $\{1, \ldots, T\}$ uniformly.

## 3 Methodology

Our GDSR consists of two components: a denoising module and a recommendation module. The former devise a dual diffusion model (DM) to denoise the original interaction graph and social graph. The latter introduces a dual graph learning framework with a diffusion-aware graph contrastive task to model user preferences based on the denoised graphs. Figure 1 shows the architecture of our GDSR.

### 3.1 Denoising Module

This subsection first introduces the denoising module's two parts: dual DMs and the cross-domain signal guidance mechanism. Next, we describe the training process of the denoising module.

*3.1.1 **Dual Diffusion Models**.* Inspired by the effectiveness of diffusion models (DMs) in data denoising [13, 35, 36, 45], we propose a dual DM structure consisting of a collaborative diffusion model (CDM) and a social diffusion model (SDM). This dual structure aims to mitigate the negative impact of irrelevant or noisy data in social recommendation. Specifically, CDM generates a denoised interaction graph $\mathcal{G_{B^*}}$ from the original interaction graph $\mathcal{G_B}$. Similarly, the SDM denoises the original social graph $\mathcal{G_S}$ to obtain a cleaner graph $\mathcal{G_{S^*}}$. Both CDM and SDM employ a forward process that gradually introduces noise to the initial graph data. Then, they utilize a reverse process to gradually recover original graphs, effectively reducing the impact of noisy signals. Next, we introduce the forward and reverse processes for our dual DM model.

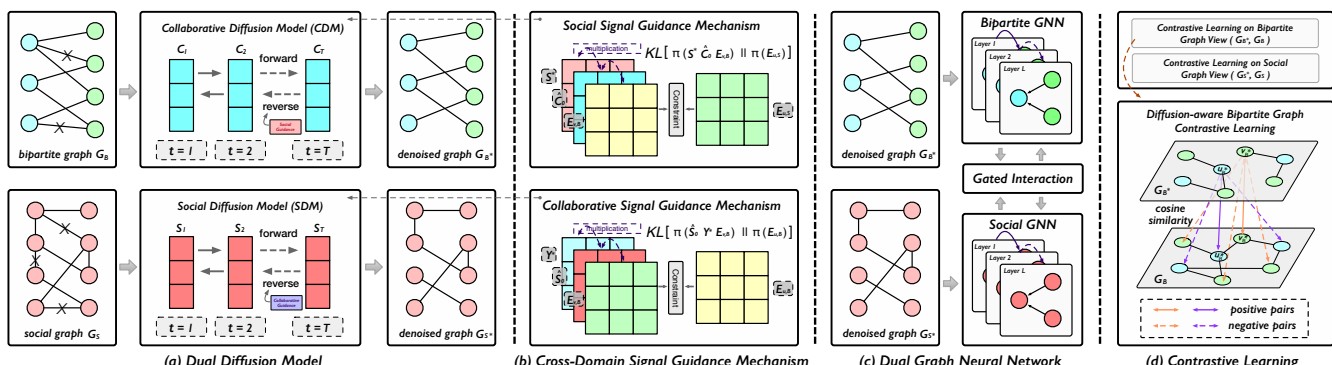

Figure 1: The structure of our GDSR, which contains a denoising module ((a)&(b)) and a recommendation module ((c)&(d)).

• **Forward process.** In CDM, we use $C_0=\{c_{u,0}\}_{u=1}^{|\mathcal{U}|}$ to denote user neighborhoods in $\mathcal{G}_\mathcal{B}$, where $c_{u,0}=[c_u^1, c_u^2, \cdots, c_u^{|\mathcal{V}|}]$ is user $u$'s neighbors over item set $\mathcal{V}$ and $c_u^i = 1$ or 0 implies whether $u$ interacts with item $i$ or not. Starting with the state $C_0$, the forward transition is performed independently on each user neighborhood as:

$$q(C_t|C_{t-1}) = \prod_{u=1}^{|\mathcal{U}|} q(c_{u,t}|c_{u,t-1}) = \prod_{u=1}^{|\mathcal{U}|} \mathcal{N}(c_{u,t}; \sqrt{1-\beta_t}c_{u,t-1}, \beta_t I), \quad (5)$$

Similarly, in SDM, user neighborhoods in $\mathcal{G}_\mathcal{S}$ is denoted as $\mathcal{S}_0 = \{s_{u,0}\}_{u=1}^{|\mathcal{U}|}$, where $s_{u,0}=[s_u^1, s_u^2, \cdots, s_u^{|\mathcal{U}|}]$ denotes user $u$'s neighbors over user set $\mathcal{U}$ and $s_u^j = 1$ or 0 indicates whether user $u$ has social relation with user $j$ or not. The forward transition in SDM is as:

$$q(\mathcal{S}_t|\mathcal{S}_{t-1}) = \prod_{u=1}^{|\mathcal{U}|} q(s_{u,t}|s_{u,t-1}) = \prod_{u=1}^{|\mathcal{U}|} \mathcal{N}(s_{u,t}; \sqrt{1-\beta_t}s_{u,t-1}, \beta_t I). \quad (6)$$

• **Reverse process.** After obtaining the noise-added user interaction neighbor $C_T$, we denoise them in the reverse process, as:

$$p_\theta(C_{t-1}|C_t) = \prod_{u=1}^{|\mathcal{U}|} p_\theta(c_{u,t-1}|C_t) = \prod_{u=1}^{|\mathcal{U}|} \mathcal{N}(c_{u,t-1}; \mu_\theta(C_t, t), \Sigma_\theta(C_t, t)).$$

where $\mu_\theta$ and $\Sigma_\theta$ denote the Gaussian parameters outputted by the our denoising neural network in CDM with parameter $\theta$.

Similarly, in SDM, the reverse process from $\mathcal{S}_T$ is defined as:

$$p_\psi(\mathcal{S}_{t-1}|\mathcal{S}_t) = \prod_{u=1}^{|\mathcal{U}|} p_\psi(s_{u,t-1}|\mathcal{S}_t) = \prod_{u=1}^{|\mathcal{U}|} \mathcal{N}(s_{u,t-1}; \mu_\psi(\mathcal{S}_t, t), \Sigma_\psi(\mathcal{S}_t, t)).$$

where $\psi$ is the parameter in SDM's denoising neural network.

• **Condition Encoders.** In the denoising process, it is crucial to use specific conditions as guidance. In the graph-based social recommendation task, in addition to the step information $t$, we introduce three conditions, including global graph semantics, personalized information, and cross-domain knowledge signals. Specifically, for CDM, the reverse process is rewritten as follows:

$$p_\theta(C_{t-1}|C_t) = \prod_{u=1}^{|\mathcal{U}|} p_\theta(c_{u,t-1}|c_{u,t}, t, g_\mathcal{B}, h_{u,\mathcal{B}}, w_{u,\mathcal{S}}), \quad (7)$$

where $g_\mathcal{B}$ is the global semantics for the bipartite graph, and we define it by applying a pooling operation to embedding matrices $\mathbf{E}_{u,\mathcal{B}}^*, \mathbf{E}_{v,\mathcal{B}}^*$ in the collaborative domain (cf. Eq.(25) in Section 3.2.1):

$$g_\mathcal{B} = \text{Pool}(\mathbf{E}_{u,\mathcal{B}}^*) + \text{Pool}(\mathbf{E}_{v,\mathcal{B}}^*). \quad (8)$$

Here, $g_\mathcal{B}$ is the graph-level property, which help in understanding the contextual semantics during the denoising process.

In Eq.(7), $h_{u,\mathcal{B}}$ represents the personalized information condition, and we directly define it using the user embedding $\mathbf{u}_\mathcal{B}^*$ from $\mathbf{E}_{u,\mathcal{B}}^*$ in the collaborative domain, as $h_{u,\mathcal{B}} = \mathbf{u}_\mathcal{B}^*$. This condition intuitively reflects user interaction behaviors with items and enables the denoising module to recognize user preferences.

For the cross-domain knowledge $w_{u,\mathcal{S}}$, it represents the semantic signals of the user in the social domain, a condition unique to the social recommendation task. According to social influence and homophily theories [30, 31], a user's preferences are influenced by those of their friends. Based on this, we define $w_{u,\mathcal{S}}$ as follows:

$$w_{u,\mathcal{S}} = \sum_{i \in \mathcal{N}_u^S} \sum_{j \in \mathcal{N}_i^\mathcal{B}} \mathbf{j}_\mathcal{B}^*. \quad (9)$$

where $\mathbf{j}_\mathcal{B}^*$ is embedding of item $j$ from $\mathbf{E}_{v,\mathcal{B}}^*$ (cf. Eq.(25)). Here, we aggregate the set of items (i.e., $\mathcal{N}_i^\mathcal{B}$) interacted with by user $u$'s social neighbors (i.e., $\mathcal{N}_u^S$) as cross-domain guidance.

Similarly, the reverse process in SDM is rewritten as follows:

$$p_\psi(\mathcal{S}_{t-1}|\mathcal{S}_t) = \prod_{u=1}^{|\mathcal{U}|} p_\psi(s_{u,t-1}|s_{u,t}, t, g_\mathcal{S}, h_{u,\mathcal{S}}, w_{u,\mathcal{B}}), \quad (10)$$

where $g_\mathcal{S}, h_{u,\mathcal{S}}, w_{u,\mathcal{B}}$ are the global graph semantics, personalized information, and cross-domain knowledge signals respectively:

$$g_\mathcal{S} = \text{Pool}(\mathbf{E}_{u,\mathcal{S}}^*), \quad h_{u,\mathcal{S}} = \mathbf{u}_\mathcal{S}^*, \quad w_{u,\mathcal{B}} = \sum_{i \in \mathcal{N}_u^\mathcal{B}} \mathbf{i}_\mathcal{B}^*, \quad (11)$$

where $\mathbf{E}_{u,\mathcal{S}}^*$ is the user social embedding matrix (cf. Eq.(25)), $\mathbf{u}_\mathcal{S}^*$ is the user social feature from $\mathbf{E}_{u,\mathcal{S}}^*$, and cross-domain signal $w_{u,\mathcal{B}}$ is obtained by aggregating the user interaction history (i.e., $\mathcal{N}_u^\mathcal{B}$).

• **Training.** To optimize our CDM and SDM, we define the following loss function according to Eq.(4), as:

$$\mathcal{L}_{\text{CDM}} = \sum_{u=1}^{|\mathcal{U}|} \mathbb{E}_t[||f_\theta(c_{u,t}, g_\mathcal{B}, h_{u,\mathcal{B}}, w_{u,\mathcal{S}}, t) - c_{u,0}||_2^2], \quad (12)$$

$$\mathcal{L}_{\text{SDM}} = \sum_{u=1}^{|\mathcal{U}|} \mathbb{E}_t[||f_\psi(s_{u,t}, g_\mathcal{S}, h_{u,\mathcal{S}}, w_{u,\mathcal{B}}, t) - s_{u,0}||_2^2]. \quad (13)$$

where $f_\theta$ and $f_\psi$ represent the denoising neural networks for CDM and SDM, respectively. We define them using a two-layer feedforward neural network. Taking $f_\theta$ as an example, its input is the concatenation of the three conditional embeddings $g_\mathcal{B}, h_{u,\mathcal{B}}$, and $w_{u,\mathcal{S}}$, along with $c_{u,t}$ and the embedding of step $t$.

*3.1.2 Denoised Graph Generation.* After training, we generate denoised interaction and social graphs $\mathcal{G}_{\mathcal{B}^*}$ and $\mathcal{G}_{\mathcal{S}^*}$. Taking $\mathcal{G}_{\mathcal{B}^*}$ as an example, for each user $u$, CDM first corrupts $c_{u,0}$ as $c_{u,0} \rightarrow c_{u,1} \cdots \rightarrow c_{u,T'}$ over $T'$ steps in the forward process. Then, CDM sets $\tilde{c}_{u,T} = c_{u,T'}$ and performs denoising as

$\tilde{c}_{u,T} \to \tilde{c}_{u,T-1} \cdots \to \tilde{c}_{u,0}$ for $T$ steps. Since the original interaction data contains noise and to preserve user preference information [45], we set $T' < T$. Next, for each user's denoised interaction, $\tilde{c}_{u,0} = [\tilde{c}_u^1, \tilde{c}_u^2, \cdots, \tilde{c}_u^{|\mathcal{V}|}]$, we design a sampling method to construct the denoised graph. Specifically, for user $u$, we first select a set $\mathcal{I}_u^c$ containing $k_c$ elements based on $\tilde{c}_{u,0}$, as follows:

$$\mathcal{I}_u^c \sim \text{Multinomial}(k_c, \pi(\tilde{c}_{u,0})), \qquad (14)$$

where $\pi$ is the softmax function and Multinomial refers to sampling $k_c$ elements based on the probability distribution $\pi(\tilde{c}_{u,0})$. We define $k_c$ as the number of neighbors user $u$ has in the original graph $\mathcal{G}_{\mathcal{B}}$.

Next, we form a candidate neighborhood from the items corresponding to the elements of set $\mathcal{I}_u^c$ in $\tilde{c}_{u,0}$ and take the intersection of this candidate neighborhood with user $u$'s original item neighborhood in $\mathcal{G}_{\mathcal{B}}$ to obtain the final neighborhood for constructing the denoised graph $\mathcal{G}_{\mathcal{B}^*}$. Similarly, we apply the same sampling strategy to construct the denoised social graph $\mathcal{G}_{\mathcal{S}^*}$. In this way, our graph generation preserves important information from the original graph while performing denoising and filtering.

### 3.1.3 *Cross-Domain Signal Guidance Mechanism*.
To further guide our dual DM in generating denoised graphs, $\mathcal{G}_{\mathcal{B}^*}$ and $\mathcal{G}_{\mathcal{S}^*}$, suitable for recommendation, we introduce a signal guidance (SG) mechanism. This method establishs communication between CDM and SDM by integrating social signals from SDM into the CDM and collaborative signals from CDM into the SDM.

Specifically, to optimize $\mathcal{G}_{\mathcal{B}^*}$, we first introduce the denoised user social relation matrix $\mathbf{S}^*$ from generated $\mathcal{G}_{\mathcal{S}^*}$ to update the predicted user-item relation matrices $\hat{C}_0$ in the CDM denoising process (by combining the $\hat{c}_0$ predicted by $f_\theta$). Then, we introduce the item feature matrix $\mathbf{E}_{v,\mathcal{B}}^*$ (*cf.* Eq.(25)) and apply the graph convolution to obtain a social-aware user feature matrix $\mathbf{E}_{c(u)}^\circ = \mathbf{S}^* \hat{C}_0 \mathbf{E}_{v,\mathcal{B}}^*$. Next, we introduce user social embeddings $\mathbf{E}_{u,\mathcal{S}}^*$ (*cf.* Eq.(25)) and utilize KL divergence as a constraint to align user embeddings in Eq.(15). This operation injects cross-domain information by aligning the distribution of user embeddings. Similarly, to optimize $\mathcal{G}_{\mathcal{S}^*}$, we introduce the denoised user-item relation matrix $\mathbf{Y}^*$ from generated $\mathcal{G}_{\mathcal{B}^*}$ and item feature matrix $\mathbf{E}_{v,\mathcal{B}}^*$ to update user relation matrix $\hat{S}_0$ in the SDM denoising process (by combining the $\hat{s}_0$ predicted by $f_\psi$) and obtain a collaborative-aware user feature matrix $\mathbf{E}_{s(u)}^\circ = \hat{S}_0 \mathbf{Y}^* \mathbf{E}_{v,\mathcal{B}}^*$. Then, based on Eq.(16), we align $\mathbf{E}_{s(u)}^\circ$ with the user collaborative feature matrix $\mathbf{E}_{u,\mathcal{B}}^*$ (*cf.* Eq.(25)).

$$\mathcal{L}_{\text{KL}(C)} = \frac{1}{|\mathcal{U}|} \Big( \pi(\mathbf{E}_{c(u)}^\circ) \cdot \big( \log \pi(\mathbf{E}_{c(u)}^\circ) - \log \pi(\mathbf{E}_{u,\mathcal{S}}^*) \big) \Big), \quad (15)$$

$$\mathcal{L}_{\text{KL}(S)} = \frac{1}{|\mathcal{U}|} \Big( \pi(\mathbf{E}_{s(u)}^\circ) \cdot \big( \log \pi(\mathbf{E}_{s(u)}^\circ) - \log \pi(\mathbf{E}_{u,\mathcal{B}}^*) \big) \Big). \quad (16)$$

In summary, the above process uses cross-domain relations and knowledge from the recommendation module to constrain $\hat{C}_0$ and $\hat{S}_0$, thereby optimizing denoised graph. In practice, we exploit batch data instead of the entire matrix to improve efficiency. In Section Sections 4.4 and 4.6.2, we validate the effectiveness of this design.

### 3.1.4 *Denoising Module Training*.
To train this module, we integrate losses from dual DM and signal guidance mechanism:

$$\mathcal{L}_{\text{Denoising}} = \mathcal{L}_{\text{CDM}} + \mathcal{L}_{\text{SDM}} + \mathcal{L}_{\text{KL}(C)} + \mathcal{L}_{\text{KL}(S)}. \quad (17)$$

## 3.2 Recommendation Module

In this section, we first introduce our dual graph neural network (GNN) for recommendation. Then, we describe the diffusion-aware contrastive learning used for model representation enhancement.

### 3.2.1 *Dual Graph Neural Networks*.
Based on the generated denoised interaction graph $\mathcal{G}_{\mathcal{B}^*}$ and social graph $\mathcal{G}_{\mathcal{S}^*}$ in Section 3.1.2, we introduce a dual GNN structure to model user preferences for items. Specifically, given a user $u_i$ and an item $v_j$, we apply a bipartite GNN (i.e., $\text{AGG}_{\mathcal{B}}$) on graph $\mathcal{G}_{\mathcal{B}^*}$ to learn their representations in the collaborative space, as follows:

$$\mathbf{u}_{i,\mathcal{B}}^l = \text{AGG}_{\mathcal{B}}^l(u_i, \mathcal{G}_{\mathcal{B}^*}) = \sum_{v_k \in \mathcal{N}_{u_i}^{\mathcal{B}}} \frac{1}{p_{\mathcal{B},ik}} \mathbf{v}_{k,\mathcal{B}}^{l-1}, \quad (18)$$

$$\mathbf{v}_{j,\mathcal{B}}^l = \text{AGG}_{\mathcal{B}}^l(v_j, \mathcal{G}_{\mathcal{B}^*}) = \sum_{u_k \in \mathcal{N}_{v_j}^{\mathcal{B}}} \frac{1}{p_{\mathcal{B},jk}} \mathbf{u}_{k,\mathcal{B}}^{l-1}, \quad (19)$$

where $p_{\mathcal{B},ik} = \sqrt{|\mathcal{N}_{u_i}^{\mathcal{B}}|}\sqrt{|\mathcal{N}_{v_k}^{\mathcal{B}}|}$, $p_{\mathcal{B},jk} = \sqrt{|\mathcal{N}_{v_j}^{\mathcal{B}}|}\sqrt{|\mathcal{N}_{u_k}^{\mathcal{B}}|}$ are normalization terms, and $\mathcal{N}_{u_i}^{\mathcal{B}}$ and $\mathcal{N}_{v_j}^{\mathcal{B}}$ are neighborhoods for $u_i$ and $v_j$ in $\mathcal{G}_{\mathcal{B}^*}$.

Additionally, we use another social GNN (i.e., $\text{AGG}_{\mathcal{S}}$) based on $\mathcal{G}_{\mathcal{S}^*}$ to learn the representation of user $u_i$ in the social space, as:

$$\mathbf{u}_{i,\mathcal{S}}^l = \text{AGG}_{\mathcal{S}}^l(u_i, \mathcal{G}_{\mathcal{S}^*}) = \sum_{u_j \in \mathcal{N}_{u_i}^{\mathcal{S}}} \frac{1}{p_{\mathcal{S},ij}} \mathbf{u}_{j,\mathcal{S}}^{l-1}, \quad (20)$$

where $p_{\mathcal{S},ij} = \sqrt{|\mathcal{N}_{u_i}^{\mathcal{S}}|}\sqrt{|\mathcal{N}_{u_j}^{\mathcal{S}}|}$ and $\mathcal{N}_{u_i}^{\mathcal{S}}$ is $u_i$'s neighborhood in $\mathcal{G}_{\mathcal{S}^*}$.

Here, the initial representations of user $u_i$ and item $v_j$ are denoted as $\mathbf{u}_i$ and $\mathbf{v}_j$, and they are used as inputs for the dual GNN (i.e., $\mathbf{u}_i = \mathbf{u}_{i,\mathcal{B}}^0 = \mathbf{u}_{i,\mathcal{S}}^0$, and $\mathbf{v}_j = \mathbf{v}_{j,\mathcal{B}}^0$). Compared to the standard GCN [19], we follow the idea in LightGCN [12] and remove the feature transformation and nonlinear activation. Other models (e.g., NGCF [46] and LR-GCCF [2]) can also be employed.

Currently, the bipartite GNN and social GNN independently model $\mathcal{G}_{\mathcal{B}^*}$ and $\mathcal{G}_{\mathcal{S}^*}$, overlooking the potential knowledge signal sharing between them. To capture the interplay between GNNs, we design a gated interaction (GI) mechanism that leverages the user as a bridge. Specifically, the GI mechanism takes user representations $\mathbf{u}_{i,\mathcal{B}}$ from $\text{AGG}_{\mathcal{B}}$ and $\mathbf{u}_{i,\mathcal{S}}$ from $\text{AGG}_{\mathcal{S}}$ at each layer as inputs, and then uses a gating mechanism to model their interactions:

$$\mathbf{u}_{i,\mathcal{B}}^{l,\diamond} = \text{GI}(\mathbf{u}_{i,\mathcal{B}}^l, \mathbf{u}_{i,\mathcal{S}}^l)[\mathcal{B}] = \mathbf{G}_{\mathcal{B}} \odot \mathbf{u}_{i,\mathcal{S}}^l + (\mathbf{1} - \mathbf{G}_{\mathcal{B}}) \odot \mathbf{u}_{i,\mathcal{B}}^l, \quad (21)$$

$$\mathbf{u}_{i,\mathcal{S}}^{l,\diamond} = \text{GI}(\mathbf{u}_{i,\mathcal{B}}^l, \mathbf{u}_{i,\mathcal{S}}^l)[\mathcal{S}] = \mathbf{G}_{\mathcal{S}} \odot \mathbf{u}_{i,\mathcal{B}}^l + (\mathbf{1} - \mathbf{G}_{\mathcal{S}}) \odot \mathbf{u}_{i,\mathcal{S}}^l, \quad (22)$$

$$\mathbf{G}_{\mathcal{B}} = \sigma\big(\mathbf{w}_{g\mathcal{B}}(\mathbf{u}_{i,\mathcal{B}}^l \oplus \mathbf{u}_{i,\mathcal{S}}^l)\big), \quad \mathbf{G}_{\mathcal{S}} = \sigma\big(\mathbf{w}_{g\mathcal{S}}(\mathbf{u}_{i,\mathcal{S}}^l \oplus \mathbf{u}_{i,\mathcal{B}}^l)\big), \quad (23)$$

where $\mathbf{u}_{i,\mathcal{B}}^{l,\diamond}, \mathbf{u}_{i,\mathcal{S}}^{l,\diamond}$ are updated embeddings, $\mathbf{G}_{\mathcal{B}}, \mathbf{G}_{\mathcal{S}}$ are gated structures, $\mathbf{w}_{g\mathcal{B}}, \mathbf{w}_{g\mathcal{S}}$ are weight matrices, $\sigma$ is the sigmoid function, and $\odot$ and $\oplus$ are the element-wise product and vector concatenation.

We integrate the GI mechanism into the current dual GNN (i.e., Eqs.(18)-(20)) to capture the interplay between GNNs, We present the matrix form of the layer-wise propagation rules, as follows:

$$\mathbf{E}_{\mathcal{B}}^l = [\mathbf{E}_{u,\mathcal{B}}^l, \mathbf{E}_{v,\mathcal{B}}^l] = (\mathbf{D}_{\mathcal{B}}^{-\frac{1}{2}} \mathbf{A}_{\mathcal{B}} \mathbf{D}_{\mathcal{B}}^{-\frac{1}{2}}) [\text{GI}(\mathbf{E}_{u,\mathcal{B}}^{l-1}, \mathbf{E}_{u,\mathcal{S}}^{l-1})[\mathcal{B}], \mathbf{E}_{v,\mathcal{B}}^{l-1}],$$

$$\mathbf{E}_{\mathcal{S}}^l = \mathbf{E}_{u,\mathcal{S}}^l = (\mathbf{D}_{\mathcal{S}}^{-\frac{1}{2}} \mathbf{A}_{\mathcal{S}} \mathbf{D}_{\mathcal{S}}^{-\frac{1}{2}}) \text{GI}(\mathbf{E}_{u,\mathcal{B}}^{l-1}, \mathbf{E}_{u,\mathcal{S}}^{l-1})[\mathcal{S}]. \quad (24)$$

where $\mathbf{D}_{\mathcal{B}}, \mathbf{D}_{\mathcal{S}}$ are diagonal matrices and $\mathbf{A}_{\mathcal{B}}, \mathbf{A}_{\mathcal{S}}$ are adjacency matrices, for corresponding denoised interaction and social graphs; $\mathbf{E}_{u,\mathcal{B}}, \mathbf{E}_{v,\mathcal{B}}$, and $\mathbf{E}_{u,\mathcal{S}}$ are user collaborative embedding matrix from $\text{AGG}_{\mathcal{B}}$, item collaborative embedding matrix from $\text{AGG}_{\mathcal{B}}$, and user social embedding matrix from $\text{AGG}_{\mathcal{S}}$, respectively.

By incorporating the GI mechanism, we facilitate the exchange of relevant information between GNN aggregations. We will analyze the efficacy of this GI mechanism in experiments (*cf.* Section 4.4).

After $L$ layers of aggregation, we combine the embeddings from each layer to form user collaborative feature $\mathbf{E}^*_{u,\mathcal{B}}$ and social feature $\mathbf{E}^*_{u,\mathcal{S}}$, and item collaborative feature $\mathbf{E}^*_{v,\mathcal{B}}$, as follows:

$$\mathbf{E}^*_{u,\mathcal{B}} = \sum_{l=0}^{L} \mathbf{E}^l_{u,\mathcal{B}}, \; \mathbf{E}^*_{u,\mathcal{S}} = \sum_{l=0}^{L} \mathbf{E}^l_{u,\mathcal{S}}, \; \mathbf{E}^*_{v,\mathcal{B}} = \sum_{l=0}^{L} \mathbf{E}^l_{v,\mathcal{B}}. \quad (25)$$

Then, we model the user $u_i$'s preference for item $v_j$, as:

$$\hat{y}_{ij} = \frac{1}{2}\left(\mathbf{u}^{*\top}_{i,\mathcal{B}}\mathbf{v}^*_{j,\mathcal{B}} + \mathbf{u}^{*\top}_{i,\mathcal{S}}\mathbf{v}^*_{j,\mathcal{B}}\right), \quad (26)$$

where $\mathbf{u}^*_{i,\mathcal{B}} = \left[\mathbf{E}^*_{u,\mathcal{B}}\right]_{u_i}, \mathbf{u}^*_{i,\mathcal{S}} = \left[\mathbf{E}^*_{u,\mathcal{S}}\right]_{u_i}, \mathbf{v}^*_{j,\mathcal{B}} = \left[\mathbf{E}^*_{v,\mathcal{B}}\right]_{v_j}$ are feature vectors.

Based on the prediction, we define the recommendation loss, as:

$$\mathcal{L}_{\text{REC}} = -\sum_{y_{ij}=1} \log(\hat{y}_{ij}) - \sum_{y_{ij}=0} \log(1-\hat{y}_{ij}) + \lambda_{\text{r}}||\Theta||^2_2, \quad (27)$$

where $\Theta$ is model parameters and $\lambda_{\text{r}}$ is the regularization strength.

*3.2.2 **Diffusion-aware Contrastive Learning**.* Recommender models using contrastive learning can effectively enhance model performance and robustness [20, 56]. We introduce a diffusion-aware contrastive learning approach. We consider the graphs after and before denoising in Section 3.1 as contrastive views (i.e., $(\mathcal{G}_{\mathcal{B}^*}, \mathcal{G}_{\mathcal{B}})$ and $(\mathcal{G}_{\mathcal{S}^*}, \mathcal{G}_{\mathcal{S}})$). We then use the dual GNN in Section 3.2.1 to process these graphs and obtain node representations. Finally, we treat the representations of the same node in different views as positive pairs and those of different nodes as negative pairs.

For users in the contrastive views $(\mathcal{G}_{\mathcal{B}^*}, \mathcal{G}_{\mathcal{B}})$, we define our contrastive loss using the InfoNCE [33], as follows:

$$\mathcal{L}_{\text{CL}^{\mathcal{B}}_u} = \sum_{u_i \in \mathcal{U}} -\log \frac{\exp\left(\cos\left(\mathbf{u}^*_{i,\mathcal{B}}, \mathbf{u}^{\star}_{i,\mathcal{B}}\right)/\tau\right)}{\sum_{u_{i'} \in \mathcal{U}} \exp\left(\cos\left(\mathbf{u}^*_{i,\mathcal{B}}, \mathbf{u}^{\star}_{i',\mathcal{B}}\right)/\tau\right)}, \quad (28)$$

where $\cos(\cdot,\cdot)$ is cosine similarity function, $\tau$ is the temperature hyper-parameter, $\mathbf{u}^*_{i,\mathcal{B}}$ and $\mathbf{u}^{\star}_{i,\mathcal{B}}$ are the representations of user $u_i$ obtained by processing $\mathcal{G}_{\mathcal{B}^*}$ and $\mathcal{G}_{\mathcal{B}}$ using our dual GNN.

Similarly, we obtain the item contrastive loss in views $(\mathcal{G}_{\mathcal{B}^*}, \mathcal{G}_{\mathcal{B}})$ as $\mathcal{L}_{\text{CL}^{\mathcal{B}}_v}$, and the user contrastive loss in views $(\mathcal{G}_{\mathcal{S}^*}, \mathcal{G}_{\mathcal{S}})$ as $\mathcal{L}_{\text{CL}^{\mathcal{S}}_u}$. Combining these terms, we get the final loss of diffusion-aware contrastive task as $\mathcal{L}_{\text{CL}} = \mathcal{L}_{\text{CL}^{\mathcal{B}}_u} + \mathcal{L}_{\text{CL}^{\mathcal{B}}_v} + \mathcal{L}_{\text{CL}^{\mathcal{S}}_v}$. The effectiveness of our contrastive learning will be validated in Sections 4.4 and 4.7.

*3.2.3 **Recommendation Module Training**.* Our recommendation module contains two components: a dual GNN and a diffusion-aware contrastive learning task. To train this module, we integrate the losses from both components as follows:

$$\mathcal{L}_{\text{Recommendation}} = \mathcal{L}_{\text{REC}} + \lambda_{\text{CL}}\mathcal{L}_{\text{CL}}, \quad (29)$$

where hyper-parameter $\lambda_{\text{CL}}$ adjust the contrastive learning strength.

## 3.3 Model Optimization

To optimize our GDSR model, we integrate the losses from both the denoising module (i.e., Eq.(17)) and the recommendation module (i.e., Eq.(29)). The combined loss is defined as $\mathcal{L}_{\text{GDSR}} = \mathcal{L}_{\text{Denoising}} + \mathcal{L}_{\text{Recommendation}}$. We then alternately train the two loss terms. The pseudocode for all the optimization procedure, including denoising and recommendation modules, is provided in **Appendix B**.

## 3.4 Model Complexity and Generalization

We introduce our GDSR's model complexity and generalization, and time analysis experiments in **Appendix C**.

## 4 Experiment

In this section, we first introduce the experimental setup and then conduct experiments to analyze the effectiveness of our GDSR (anonymous code https://anonymous.4open.science/r/GDSR-WWW).

## 4.1 Experimental Setup

This subsection introduces the datasets, comparison methods, hyper-parameter settings, and evaluation metrics.

*4.1.1 **Datasets**.* We use three real-world datasets: Yelp, Douban, and Flixster, which are collected from online applications and widely used in social recommendation. Each dataset contains the user-item interaction and user social information. More details of the datasets are in **Appendix D.1**. For each dataset, we select 60%, 20%, and 20% of interactions as training, evaluation, and test sets, respectively. Table 1 shows the statistics of the three datasets.

**Table 1: Statistical details of the three datasets.**

| Datasets | # users | # items | # interactions | # social links |
|----------|---------|---------|----------------|----------------|
| Yelp | 16,239 | 14,284 | 198,397 | 158,590 |
| Douban | 2,848 | 39,586 | 894,887 | 35,770 |
| Flixster | 42,935 | 15,816 | 2,448,110 | 517,966 |

*4.1.2 **Comparison Methods**.* We compare GDSR with four group methods: (1) graph-based collaborative recommenders (i.e., LR-GCCF [2] and LightGCN [12]), (2) graph-based social recommenders (i.e., GraphRec+ [7] and DiffNet++ [49]), (3) denoising graph-based collaborative recommenders (i.e., RGCF [40], DDRM [58], AdaGCL [17]), and (4) denoising graph-based social recommenders (i.e., GDMSR [34], DSL [43], GDSSL [20], and RecDiff [22]). The characteristics of these baselines are introduced in **Appendix D.2**.

*4.1.3 **Hyper-parameter Settings**.* For baselines, we implement them based on their source code. More about the implementation details are in **Appendix D.3**. For our GDSR, diffusion step $T$ is tuned in $\{2, 5, 10, 20, 50, 100\}$. For the noise, its scale $s$, lower bound $\alpha_{\min}$, and upper bound $\alpha_{\max}$ are searched in $\{10^{-5}, 10^{-4}, 10^{-3}, 10^{-2}, 10^{-1}\}$, $\{10^{-4}, 10^{-3}, 2 \times 10^{-3}\}$, and $\{5 \times 10^{-3}, 10^{-2}, 2 \times 10^{-2}\}$. In the dual GNN, we set the embedding size as 32 and the aggregation layer as 2. We study the impact of key hyper-parameters in Section 4.5.

*4.1.4 **Evaluation Metrics**.* Two scenarios are used to evaluate the model performance: (1) in the top-k recommendation, precision (P@K) and recall (R@K) are utilized as the metrics, where K is set as 10 by default; (2) in click-through rate (CTR) prediction, area under curve (AUC) and accuracy (ACC) are adopted as the metrics.

## 4.2 Performance Comparison

Tables 2 and 3 present the model performance on the three datasets. We find that: (1) Graph-based social recommenders generally outperform graph-based collaborative recommenders due to the use of additional social information. However, this is not always the case. For example, LightGCN sometimes outperforms DiffNet++ and GraphRec+ on the Yelp and Douban datasets. This suggests that

social connections might contain noise or irrelevant information. (2) Denoising collaborative and social recommenders outperform their respective non-denoising graph-based methods. This further indicates that irrelevant information in user-item interactions and social relations could affect user preference modeling. It also highlights the necessity of denoising both interaction and social data. (3) Our GDSR shows superior performance, indicating the effectiveness of its denoising and recommendation modules. Specifically, in the top-K recommendation, GDSR outperforms the best baselines with a 4.87%, 2.12%, and 1.48% increase in P@10 on the three datasets, respectively. For CTR prediction, our GDSR achieves average AUC gains of 2.80%, 2.03%, and 1.88% on the three datasets. We will discuss the effectiveness of key designs of GDSR in Section 4.4.

**Table 2: Top-k recommendation results. * denotes the statistical significance for $p < 0.001$ compared to the best baseline.**

| Method | Yelp | | Douban | | Flixster | |
|---|---|---|---|---|---|---|
| | P@10 | R@10 | P@10 | R@10 | P@10 | R@10 |
| LR-GCCF | 0.0162 | 0.0389 | 0.1823 | 0.0655 | 0.1051 | 0.1196 |
| LightGCN | 0.0178 | 0.0415 | 0.1969 | 0.0684 | 0.1073 | 0.1224 |
| GraphRec+ | 0.0170 | 0.0394 | 0.1895 | 0.0662 | 0.1109 | 0.1263 |
| DiffNet++ | 0.0174 | 0.0409 | 0.1998 | 0.0703 | 0.1087 | 0.1260 |
| RGCF | 0.0182 | 0.0436 | 0.2031 | 0.0697 | 0.1129 | 0.1238 |
| DDRM | 0.0193 | 0.0471 | 0.2110 | 0.0750 | 0.1186 | 0.1289 |
| AdaGCL | 0.0185 | 0.0442 | 0.2114 | 0.0761 | 0.1160 | 0.1278 |
| GDMSR | 0.0179 | 0.0420 | 0.2007 | 0.0708 | 0.1122 | 0.1265 |
| DSL | 0.0195 | 0.0458 | 0.2059 | 0.0736 | 0.1174 | 0.1277 |
| GDSSL | 0.0176 | 0.0416 | 0.2028 | 0.0717 | 0.1180 | 0.1286 |
| RecDiff | 0.0187 | 0.0423 | 0.2086 | 0.0749 | 0.1195 | 0.1310 |
| GDSR | **0.0206*** | **0.0497*** | **0.2162*** | **0.0781*** | **0.1217*** | **0.1383*** |

**Table 3: CTR prediction results. * denotes the statistical significance for $p < 0.001$ compared to the best baseline.**

| Method | Yelp | | Douban | | Flixster | |
|---|---|---|---|---|---|---|
| | AUC | ACC | AUC | ACC | AUC | ACC |
| LR-GCCF | 0.7658 | 0.7066 | 0.8567 | 0.7869 | 0.9353 | 0.8844 |
| LightGCN | 0.7820 | 0.7209 | 0.8596 | 0.7964 | 0.9387 | 0.8853 |
| GraphRec+ | 0.7704 | 0.7110 | 0.8609 | 0.7933 | 0.9472 | 0.8919 |
| DiffNet++ | 0.7881 | 0.7245 | 0.8710 | 0.7976 | 0.9423 | 0.8928 |
| RGCF | 0.8011 | 0.7388 | 0.8655 | 0.7950 | 0.9436 | 0.8922 |
| DDRM | 0.8093 | 0.7492 | 0.8784 | 0.8015 | 0.9501 | 0.8937 |
| AdaGCL | 0.8067 | 0.7397 | 0.8812 | 0.8101 | 0.9589 | 0.9005 |
| GDMSR | 0.7996 | 0.7303 | 0.8732 | 0.7998 | 0.9490 | 0.8934 |
| DSL | 0.7933 | 0.7275 | 0.8798 | 0.8072 | 0.9517 | 0.8960 |
| GDSSL | 0.7950 | 0.7286 | 0.8763 | 0.8009 | 0.9544 | 0.8968 |
| RecDiff | 0.8048 | 0.7314 | 0.8801 | 0.8045 | 0.9605 | 0.9096 |
| GDSR | **0.8160*** | **0.7520*** | **0.8894*** | **0.8182*** | **0.9669*** | **0.9156*** |

## 4.3 Performance w.r.t Sparsity Degrees

This subsection studies the performance of GDSR and several representative baselines (i.e., DiffNet++ [49], AdaGCL [17], and RecDiff [22]) in handling user behavior with varying sparsity levels. Following [17, 46, 49], we group users based on the number of interactions they have in the training set and then evaluate the performance of these user groups in the test set. Specifically, we divide users into three groups while trying to maintain a similar total number of interactions for each group in the test set. We label these groups based on user interaction density, from low to high, as Group 1, Group 2, and Group 3. Figure 2 shows the P@10 results. We find that as interaction density increases, the accuracy of models improves,

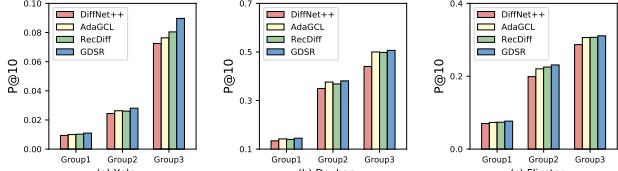

**Figure 2: Performance under different sparsity groups.**

indicating that high-quality recommendation requires enough user-item interactions. Moreover, our GDSR achieves the best results, showing its robust performance in different levels of data sparsity.

## 4.4 Ablation Study

In this subsection, we conduct ablation experiments to analyze the key design elements of our GDSR model in detail.

### 4.4.1 *Effect of Model Components*. The denoising and the recommendation modules are two crucial components of our GDSR. To analyze our denoising module, we consider four operations:

- w/o CDM: Removing the collaborative diffusion model (CDM).
- w/o SDM: Removing the social diffusion model (SDM).
- w/o dual DM: Removing the dual diffusion model (dual DM).
- w/o SG: Removing the signal guidance (SG) mechanism.

For the recommendation module, we consider two operations:

- w/o GI: Removing the gated interaction (GI) mechanism.
- w/o DCL: Removing diffusion-aware contrastive learning (DCL).

Table 4 shows the results for P@10 and AUC. From these results, we can draw the following conclusions: (1) For the denoising module, removing the CDM and/or SDM from the dual DM decreases model performance. This indicates the presence of noise in interaction and social graph data, and demonstrates that our dual DM improves model performance by denoising the data. Furthermore, the results validate the effectiveness of the SG mechanism, which guides the dual DM to better denoise the data. (2) For the recommendation module, the lower performance after removing the GI module highlights the importance of modeling information interaction between our dual GNNs. Additionally, the ablation of DGL shows that our diffusion-enhanced data augmentation strategy is crucial for improving model performance. (3) In general, removing any operation from our GDSR method reduces its performance, showing the soundness and effectiveness of our model design.

**Table 4: Ablation study of key designs in our GDSR.**

| Operation | Yelp | | Douban | | Flixster | |
|---|---|---|---|---|---|---|
| | P@10 | AUC | P@10 | AUC | P@10 | AUC |
| w/o CDM | 0.0197 | 0.8065 | 0.2096 | 0.8822 | 0.1205 | 0.9615 |
| w/o SDM | 0.0201 | 0.8104 | 0.2130 | 0.8858 | 0.1193 | 0.9601 |
| w/o dual DM | 0.0182 | 0.7944 | 0.2021 | 0.8784 | 0.1128 | 0.9525 |
| w/o SG | 0.0199 | 0.8094 | 0.2144 | 0.8887 | 0.1205 | 0.9653 |
| w/o GI | 0.0202 | 0.8123 | 0.2144 | 0.8882 | 0.1206 | 0.9640 |
| w/o DCL | 0.0195 | 0.8112 | 0.2137 | 0.8873 | 0.1198 | 0.9624 |
| GDSR | **0.0206** | **0.8160** | **0.2162** | **0.8894** | **0.1217** | **0.9669** |

### 4.4.2 *Plug-In Effect of Denoising Module*. We further analyze our denoising module. This module can be seen as a **plug-and-play** component for denoising user-item interaction graphs and user-user social graphs, thereby enhancing recommendation

performance. To validate this plug-and-play nature, we integrate our denoising module into two graph-based social recommenders DiffNet++ [49] and GraphRec+ [7]. Two methods use both the interaction and social graphs for recommendation. Figure 3 shows the performance of these two methods, both with and without our denoising module, on the three datasets. The results show that our proposed denoising module consistently improves the performance of two different base models, further validating its effectiveness.

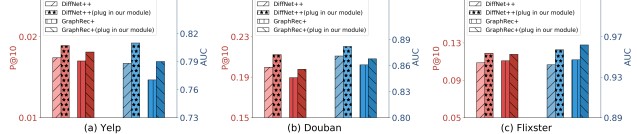

**Figure 3: Effect of our denoising module on two different backbones. In each subfigure, P@10 results (red bars) on the left, while AUC results (blue bars) on the right.**

## 4.5 Hyper-parameter Sensitivity Analysis

In this section, we analyze several important hyper-parameters in our GDSR. Specifically, we study diffusion step $T$ and noise scale $s$ in the denoising module, and dual GNN layer size $L$ and diffusion-aware contrastive learning strength $\lambda_{CL}$ in the recommendation module. The results on the Douban dataset are shown in Figure 4. The observations can be summarized as: (1) Increasing the diffusion step $T$ initially improves performance, but then it leads to a decrease. In addition, since a larger $T$ increases time costs, we select $T = 10$ to achieve a balance between nice performance and low costs. (2) As the noise scale $s$ increases, performance improves initially when compared to training without noise, demonstrating the benefits of graph denoising optimization. However, an excessively high noise scale adversely affects performance. Therefore, it is crucial to choose a relatively small noise scale $s = 10^{-4}$. (3) Increasing the number of layer $L$ in dual GNN enhances performance to a certain extent, but too many layers will increase the time complexity. In our experiments, setting $L = 2$ is a nice choice. (4) For the contrastive learning strength $\lambda_{CL}$, setting it too high can cause the model to overemphasize the contrastive task, which reduces performance. Conversely, a too-low value may not provide sufficient self-supervised signals. The nice performance is typically achieved with $\lambda_{CL}$ values between $10^{-3}$ and $10^{-2}$.

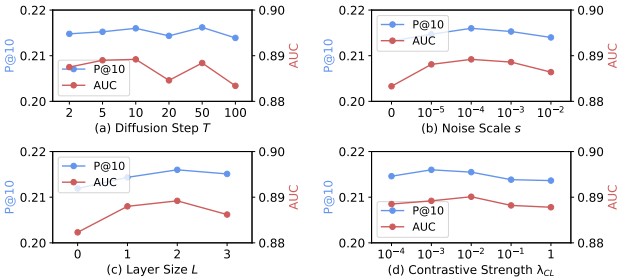

**Figure 4: Hyper-parameter sensitivity analysis.**

## 4.6 Anti-Noise Capacity Analysis

We first study the robustness of our GDSR to noisy interaction and social data. Then, we analyze the denoising effect of GDSR on several specific cases of user-item interactions and social relations.

*4.6.1* **Performance w.r.t. Data Noise Degree.** In this subsection, we analyze the anti-noise capacity of our GDSR. Specifically, following [17, 22, 40, 43], we replace a certain proportion (i.e., 10% and 20%) of user-item interactions/user-user social relations with noise signals in the interaction graph and social graph, while keeping the test set unchanged. For experiments with noise added to interactions, we compare our GDSR with denoising collaborative recommenders (i.e., RGCF [40] and AdaGCL [17]). For experiments with noise added to social data, we compare our model with denoising social recommenders (i.e., DSL [43] and RecDiff [22]). Additionally, we also include DiffNet++ [49] as a baseline. Figure 5 shows the results on the Douban dataset. Similar results are observed on other datasets, which are not elaborated here due to space limitations. From the results, we find that: (1) Adding noise to the data affects model performance. Moreover, adding noise to interaction data impacts the model more than adding noise to social data, indicating that the model is more sensitive to noise in interaction data in social recommendations. (2) In scenarios where noise is added to user-item interactions, the performance of DiffNet++ declines more than that of RGCF, AdaGCL and, our GDSR. Similarly, when noise is added to social relations, the performance decline of DSL, RecDiff, and GDSR is less than that of DiffNet++. This highlights the importance of anti-noise strategies in maintaining performance. (3) Compared to denoising baselines, our GDSR shows the smallest performance drop, indicating that our denoising strategy effectively mitigates the impact of noise in the interaction and social data.

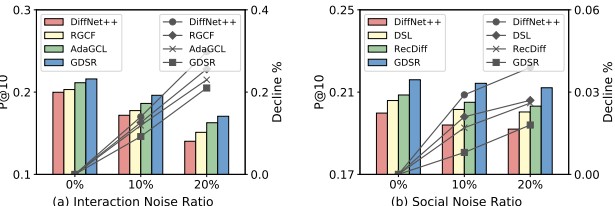

**Figure 5: Model performance w.r.t. interaction and social noise ratio. The bar represents P@10 results and the line represents the percentage of performance degradation.**

*4.6.2* **Case Study.** We aim to further investigate the data denoising of our model through a case study. Specifically, in the Yelp dataset, we randomly select two pairs of user-item interactions (i.e., $(u_{2865}, i_{4298})$ and $(u_{4964}, i_{2288})$) predicted as noise by our model, and two pairs of user-user social relations (i.e., $(u_{1982}, i_{9317})$ and $(u_{6115}, i_{5443})$) also predicted as noise. We introduce the item attribute information (i.e., category and city) to assist in our case study, as shown in Figure 6. Note that these attributes are **not** provided to our model during training, and they are used here solely for the case study. Figure 6 (a) shows the denoising of user-item interactions. Based on item attributes and user social neighbors, we can rank the preferences of a user's social neighbors for item attributes. We find that the reason for denoising the interaction $(u_{2865}, i_{4298})$ may be because item $i_{4298}$ does not match user $i_{2865}$'s social neighbors' preferences. The same reasoning applies to the interaction $(u_{4964}, i_{2288})$. Figure 6 (b) shows the denoising of social information. We utilize the user's historical preference for item attributes for analysis. The reason for denoising the social relations $(u_{1982}, i_{9317})$ and $(u_{6115}, i_{5443})$ may be due to the significant difference in item attribute preferences between the users in each pair.

The above analysis not only reflects the rationale behind our denoising method but also shows that our denoising neural network (i.e., $f_\theta$ and $f_\psi$) and SG mechanism (i.e., Eqs.(15) and (16)) can inject cross-domain signals into the data denoising process.

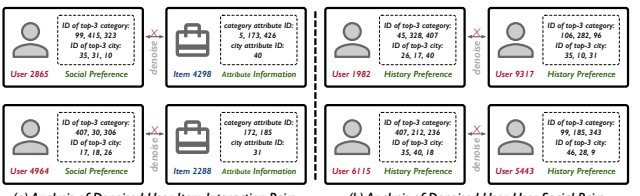

(a) Analysis of Denoised User-Item Interaction Pairs  (b) Analysis of Denoised User-User Social Pairs

**Figure 6: Case study for our denoising module.**

### 4.7 Embedding Analysis

In this subsection, we analyze the embeddings generated by our GDSR. Following [23, 59], we plot the embedding distributions of items from the Douban dataset utilizing Gaussian kernel density estimation (KDE) in a 2-dimensional space. We also compare our GDSR with two social recommender baselines DiffNet++ [49] and RecDiff [22]. The results are shown in Figure 7. We find that compared to the two baselines, the embeddings learned by our GDSR are relatively more uniformly distributed. Based on prior research [42, 59], we believe this result shows the advantage of our model in modeling the data feature diversity. We attribute this advantage to our diffusion-aware contrastive learning (i.e., Section 3.2.2). To validate this, we also plot item embeddings generated by GDSR$_{\text{w/o DCL}}$, which removes diffusion-aware contrastive learning from GDSR. We observe that the item embedding distribution of GDSR$_{\text{w/o DCL}}$ is less uniform compared to GDSR. This indicates that our contrastive learning enhances the model representation learning.

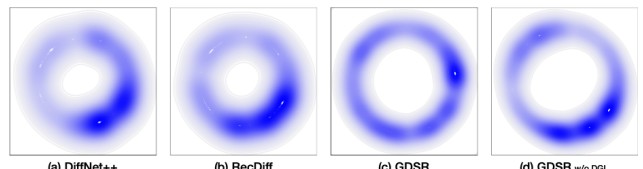

(a) DiffNet++  (b) RecDiff  (c) GDSR  (d) GDSR$_{\text{w/o DGL}}$

**Figure 7: Embedding analysis for DiffNet++, RecDiff, GDSR.**

## 5 Related Work

This section reviews two relevant research areas: social and denoising recommendation. We discuss another area, contrastive learning in recommendation, in **Appendix E** due to the space limitation.

### 5.1 Social Recommendation

Social recommendation utilizes social information to improve recommendation performance [39]. Prior research primarily focuses on employing matrix factorization techniques (e.g., ensemble [27], co-factorization [11], and regularization [28]) for social recommendation. With the development of deep learning, researchers employ diverse neural components (e.g., attention networks [1], multilayer perceptrons [5], and recurrent neural networks [38]) to design social recommenders. Recently, modeling interaction and social data from a graph perspective has gained traction. These studies utilize graph neural networks to enhance representation learning by considering interaction and social graph structures [7, 8, 45]. However, most

social recommenders overlook the handling of noise in user-item interactions and social data. To this end, inspired by the success of the diffusion model (DM) in denoising tasks [25, 35, 36, 41], we design a dual DM to generate denoised user-item interaction and social graphs. The results show that our denoising method effectively denoises the recommendation data and improves performance.

### 5.2 Denoising Recommendation

The performance of recommender models could be limited by the presence of redundant and noisy data [24, 32, 57]. To address this issue, current methods are primarily designed from data cleaning and model perspectives [58]. Data cleaning-based methods often rely on specific heuristic assumptions to remove noisy data [3, 4] or assign lower weights to potentially noisy data [34, 44, 47]. However, this approach reduces the adaptability when the dataset or backend model changes. Model perspective methods focus on enhancing the noise resistance of recommenders [9, 43, 48, 51]. However, they typically depend on a single model to convert noisy data into clean data, which makes it challenging to identify noise patterns effectively and places high demands on the model representation capacity. In this paper, we employ the multi-step denoising idea of diffusion models (DMs) to denoise data. In recent years, several approaches explore the use of DMs in recommendation. For example, DiffRec [45] and DDRM [58] integrate DMs into the modeling of user-item interactions. However, they focus solely on the collaborative filtering task. RecDiff [22] and GDSSL [20] use DMs to mitigate social noise. Despite their effectiveness, we believe our GDSR differs from theirs in two key aspects: First, their methods do not incorporate condition guidance relevant to the social recommendation task in their DM design, and they lack specific adaptations tailored to this task. In contrast, our GDSR employs a dual DM specifically designed and customized for social recommendation, incorporating cross-domain guidance strategies (i.e., denoising neural network and signal guidance mechanism) to more effectively guide the denoising process. Second, both methods overlook the noise problem in the interaction graph, while our GDSR mitigates noise in both interaction and social graphs, establishing mutual collaboration between the denoising processes of the two graphs. Experimental results further show that our GDSR outperforms these methods.

## 6 Conclusion

In this paper, we propose a graph-based social recommender GDSR, which consists of two key steps: graph denoising and recommendation prediction. Our GDSR first leverages dual diffusion models to denoise the user-item interaction and user-user social graphs. To guide our dual diffusion models, we design a denoising neural network and a signal guidance mechanism, both of which inject cross-domain knowledge signals into the diffusion process. Then, based on the denoised graphs, our GDSR introduces a dual graph learning structure to learn user and item representations for recommendation. To enhance the robustness of the model representation, we introduce a diffusion-aware graph contrastive learning task. Experiment results on three real-world datasets show that our GDSR outperforms several state-of-the-art recommender baselines. For future work, we plan to design the latent space diffusion strategy and acceleration algorithm to improve our GDSR's training efficiency.

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

# Appendix

In this appendix, we provide additional details omitted from the main paper due to the space limitation. We begin with a table of key notations used in this paper (Appendix A), followed by the optimization processes (Appendix B), and a complexity analysis and generalization discussion (Appendix C) of GDSR. Next, we describe experiment settings in detail (Appendix D). Finally, we introduce supplementary discussions on the related work (Appendix E).

## A. Notation Table

The notations used in this paper are summarized in Table 5.

**Table 5: Summary of key notations.**

| Symbol | Explanation |
|---|---|
| $\mathcal{U}, \mathcal{V}$ | user set and item set |
| $\mathbf{Y}, \mathbf{S}$ | interaction matrix and social matrix |
| $\mathcal{G}_{\mathcal{B}}, \mathcal{G}_{\mathcal{S}}$ | original interaction graph and social graph |
| $\mathcal{G}_{\mathcal{B}^*}, \mathcal{G}_{\mathcal{S}^*}$ | denoised interaction graph and social graph |
| $T, s$ | diffusion step and noise scale |
| $\mathbf{u}_i, \mathbf{v}_j$ | initial embedding for user $u_i \in \mathcal{U}$ and item $v_j \in \mathcal{V}$ |
| $\mathbf{E}_{u,\mathcal{B}}^*, \mathbf{E}_{v,\mathcal{B}}^*$ | user and item collaborative embedding matrices |
| $\mathbf{E}_{u,\mathcal{S}}^*$ | user social embedding matrix |
| $\mathrm{AGG}_{\mathcal{B}}$ | Neighborhood aggregation function for graph $\mathcal{G}_{\mathcal{B}^*}$ |
| $\mathrm{AGG}_{\mathcal{S}}$ | Neighborhood aggregation function for graph $\mathcal{G}_{\mathcal{S}^*}$ |
| $\hat{y}_{ij}$ | Predicted user $u_i$'s preference for item $v_j$ |

## B. Optimization of Our GDSR

To optimize our GDSR model, we integrate the losses from both the denoising module (i.e., Eq.(17)) and the recommendation module (i.e., Eq.(29)). The combined loss is defined as follows:

$$\mathcal{L}_{\mathrm{GDSR}} = \mathcal{L}_{\mathrm{Denoising}} + \mathcal{L}_{\mathrm{Recommendation}}. \tag{30}$$

We then alternately train the losses of two modules. The pseudocodes of our GDSR are introduced in Algorithms 1-4.

---

**Algorithm 1:** Optimization procedure of GDSR

**Input:** User and item sets $\mathcal{U}$ and $\mathcal{V}$, interaction matrix $\mathbf{Y}$, original interaction graph $\mathcal{G}_{\mathcal{B}}$, original social graph $\mathcal{G}_{\mathcal{S}}$, diffusion step $T$, inference step $T'$, $f_\theta$ in CDM, $f_\psi$ in SDM, recommendation module parameter $\Theta$, numbers of training $\Gamma_{\mathrm{de}}$ and $\Gamma_{\mathrm{rec}}$.

1 **for** *number of iterations for GDSR* **do**
2    **for** *number of training $\Gamma_{de}$ for denoising* **do**
3      Train the denoising module (Algorithm 2);
4    **end**
5    Generate denoised graphs (Algorithm 3);
6    **for** *number of training $\Gamma_{rec}$ for recommendation* **do**
7      Train the recommendation module (Algorithm 4);
8    **end**
9 **end**
**Output:** User preference prediction function $\mathcal{F}$.

---

Algorithm 1 alternately optimizes the denoising module and the recommendation module. Specifically, in each outer loop, we first train the denoising module using Algorithm 2, then generate the denoised graph using Algorithm 3, and finally train the recommendation model using Algorithm 4. In this paper, we set the number of training iterations for both modules as $\Gamma_{\mathrm{de}} = \Gamma_{\mathrm{rec}} = 1$.

---

**Algorithm 2:** Denoising Module Training of GDSR

**Input:** Original interaction graph $\mathcal{G}_{\mathcal{B}}$, original social graph $\mathcal{G}_{\mathcal{S}}$, diffusion step $T$, $f_\theta$ in CDM, $f_\psi$ in SDM, user collaborative feature $\mathbf{E}_{u,\mathcal{B}}^*$, user social feature $\mathbf{E}_{u,\mathcal{S}}^*$, item collaborative feature $\mathbf{E}_{v,\mathcal{B}}^*$.

1 Sample a user-item interaction batch $C_u$ from $\mathcal{G}_{\mathcal{B}}$;
2 Sample a user-user interaction batch $\mathcal{S}_u$ from $\mathcal{G}_{\mathcal{S}}$;
3 Sample $t \sim \mathbf{U}(1, T)$;
4 Compute $C_{u,t}$ given $C_{u,t-1}$ and $t$ via $q(C_{u,t}|C_{u,t-1})$;
5 Compute $\mathcal{S}_{u,t}$ given $\mathcal{S}_{u,t-1}$ and $t$ via $q(\mathcal{S}_{u,t}|\mathcal{S}_{u,t-1})$;
6 Calculate $\mathcal{L}_{\mathrm{Denoising}}$ by Eq.(17);
7 Take gradient descent step on $\nabla_\theta(\mathcal{L}_{\mathrm{Denoising}})$ and $\nabla_\psi(\mathcal{L}_{\mathrm{Denoising}})$ to optimize parameters $\theta$ and $\psi$;
**Output:** Optimized $f_\theta$ and optimized $f_\psi$.

---

**Algorithm 3:** Denoised Graph Generation

**Input:** User set $\mathcal{U}$, original interaction graph $\mathcal{G}_{\mathcal{B}}$, original social graph $\mathcal{G}_{\mathcal{S}}$, diffusion step $T$, inference step $T'$, optimized $f_\theta$ in CDM, optimized $f_\psi$ in SDM, user collaborative feature $\mathbf{E}_{u,\mathcal{B}}^*$, user social feature $\mathbf{E}_{u,\mathcal{S}}^*$.

1 **forall** $u \in \mathcal{U}$ **do**
2    Sample $\boldsymbol{\epsilon} \sim \mathcal{N}(\mathbf{0}, \boldsymbol{I})$;
3    Compute $\boldsymbol{c}_{u,T'}$ given $\boldsymbol{c}_{u,0}, T', \boldsymbol{\epsilon}$, and set $\tilde{\boldsymbol{c}}_{u,T} = \boldsymbol{c}_{u,T'}$;
4    Compute $\boldsymbol{s}_{u,T'}$ given $\boldsymbol{s}_{u,0}, T', \boldsymbol{\epsilon}$, and set $\tilde{\boldsymbol{s}}_{u,T} = \boldsymbol{s}_{u,T'}$;
5    **for** $t = T, \dots, 1$ **do**
6      Compute $\tilde{\boldsymbol{c}}_{u,t-1}$ from $\tilde{\boldsymbol{c}}_{u,t}$ via $f_\theta$
7      Compute $\tilde{\boldsymbol{s}}_{u,t-1}$ from $\tilde{\boldsymbol{s}}_{u,t}$ via $f_\psi$;
8    **end**
9    Construct a set of the user denoised interaction neighborhood based on $\tilde{\boldsymbol{c}}_{u,0}$;
10    Construct a set of the user denoised social neighborhood based on $\tilde{\boldsymbol{s}}_{u,0}$;
11 **end**
12 Construct denoised interaction graph $\mathcal{G}_{\mathcal{B}^*}$ and denoised social graph $\mathcal{G}_{\mathcal{S}^*}$ based on user denoised neighbor sets;
**Output:** Graph $\mathcal{G}_{\mathcal{B}^*}$ and graph $\mathcal{G}_{\mathcal{S}^*}$.

---

## C. Model Complexity and Generalization

In this section, we first analyze the model complexity of our GDSR in terms of model size and time complexity. Then, we discuss the model generalization of our GDSR.

***C.1 Model Size.*** The model parameter size of our GDSR is from two parts: (1) For the denoising module, it uses $O((2 \cdot |\mathcal{U}|) \cdot d_{\mathrm{dm}})$ parameters, where $d_{\mathrm{dm}}$ is the hidden space dimension for the dual DM. In practice, a smaller $d_{\mathrm{dm}}$ (e.g., 32) makes our method achieve

nice performance. (2) For the recommendation module, it requires $O((|\mathcal{U}| + |\mathcal{V}|) \cdot d_{\text{gnn}})$ parameters for user and item embedding matrices and $O(L \cdot d_{\text{gnn}}^2)$ parameters for the weight matrices in the dual GNN (i.e., from the GI mechanism), where $L$ is the layer number and $d_{\text{gnn}}$ is the embedding size in dual GNN. Compared with embedding matrices, weight parameters are lighter and can be neglected. Generally, the parameter size of the recommendation module aligns with many GNN-based social recommenders.

---

**Algorithm 4:** Recommendation Module Training of GDSR

**Input:** Interaction matrix $\mathbf{Y}$, denoised graphs $\mathcal{G}_{\mathcal{B}^*}$, $\mathcal{G}_{\mathcal{S}^*}$.

1   Initialize all the parameter $\Theta$ in recommendation module;
2   Draw a batch of interaction data $\mathbf{Y}_b$ from $\mathbf{Y}$;
3   **forall** $(u, v) \in \mathbf{Y}_b$ **do**
4     Calculate user and item embeddings (i.e., $\mathbf{E}_{u,\mathcal{B}}^*$, $\mathbf{E}_{u,\mathcal{S}}^*$, $\mathbf{E}_{v,\mathcal{B}}^*$) based on $\mathcal{G}_{\mathcal{B}^*}$, $\mathcal{G}_{\mathcal{S}^*}$ via Eqs.(24) and (25);
5     Calculate the user-item interaction prediction (i.e., $\hat{y}_{ij}$ for user $u_i$ and item $v_j$) according to Eq.(26);
6     Calculate loss $\mathcal{L}_{\text{Recommendation}}$ according to Eq.(29) ;
7     Take gradient descent step on $\nabla_\Theta(\mathcal{L}_{\text{Recommendation}})$;
8   **end**

**Output:** Prediction function $\mathcal{F}(u, v|\Theta, \mathbf{Y}, \mathcal{G}_{\mathcal{B}^*}, \mathcal{G}_{\mathcal{S}^*})$, user collaborative features $\mathbf{E}_{u,\mathcal{B}}^*$, user social features $\mathbf{E}_{u,\mathcal{S}}^*$, item collaborative features $\mathbf{E}_{v,\mathcal{B}}^*$.

---

**C.2 Time Complexity**. The time complexity of GDSR comes from two parts. (1) In the denoising module, our dual DM takes $O(b \cdot d_{\text{dm}} \cdot |\mathcal{U}|)$ time for training, where $b$ is the batch size. In addition, our SG mechanism introduces $O(b^2 \cdot |\mathcal{U}|)$ time for training. (2) In the recommendation module, the time complexity of our dual GNN is $O((|\mathcal{E}_{\mathcal{B}^*}| + |\mathcal{E}_{\mathcal{S}^*}|) \cdot (L \cdot d_{\text{gnn}} + L \cdot d_{\text{gnn}}^2))$, where $\mathcal{E}_{\mathcal{B}^*}$ and $\mathcal{E}_{\mathcal{S}^*}$ are the edge sets in denoised bipartite and social graphs. For the diffusion-aware contrastive task, the time cost is $O(b \cdot (|\mathcal{E}_{\mathcal{B}^*}| + |\mathcal{E}_{\mathcal{S}^*}|) \cdot d_{\text{gnn}})$.

**C.3 Time Efficiency Analysis**. In this section, we conduct experiments to further analyze the time efficiency of our GDSR. Following [22], we record the time cost per epoch during training and the inference time during testing. We also compare our GDSR with three representative recommender baselines (i.e., DiffNet++ [49], AdaGCL [17], and RecDiff [22]). All experiments are conducted on a Linux server with an Intel(R) Xeon(R) Gold CPU 5218@2.3GHz and a Quadro RTX 5000 GPU. The experiment results are shown in Table 5. Generally, during the training phase, we find that AdaGCL, RecDiff, and GDSR introduce additional time costs for denoising operations compared to traditional graph-based social recommenders DiffNet++, which is because denoising operation requires additional time costs. During the testing phase, the inference time required by the models is similar. Generally, we consider the time costs of our GDSR to be acceptable. In the future, we plan to design the latent space diffusion strategy and acceleration algorithms to further improve the training efficiency of our GDSR.

**C.4 Model Generalization**. In this paper, we propose a social recommendation model GDSR, which includes a denoising module and a recommendation module. GDSR first denoises the interaction and social graphs, and then generates recommendations based on

**Table 6: Running time efficiency analysis about the training time per epoch and testing time (s: second).**

| Method | Yelp | | Douban | | Flixster | |
|---|---|---|---|---|---|---|
| | Train | Test | Train | Test | Train | Test |
| DiffNet++ | 4.552s | 1.095s | 26.553s | 8.832s | 75.603s | 41.847s |
| AdaGCL | 102.480s | 1.028s | 203.869s | 7.696s | 1175.386s | 36.756s |
| RecDiff | 9.972s | 0.881s | 31.279s | 7.314s | 103.485s | 35.555s |
| GDSR | 16.911s | 0.980s | 78.674s | 9.224s | 451.197s | 40.503s |

the denoised graphs. Although we introduce a specific recommendation module, we believe that the denoising module in GDSR does not rely on a particular backend recommender model. It can be integrated into other social recommenders to enhance performance. For example, experiments in Section 4.4.2 show that after incorporating our denoising module, both GraphRec+ [7] and DiffNet++ [49] backbones achieve improved results, demonstrating the generalization capability of the denoising module. Beyond the backbone model, we also believe that the proposed idea of the dual graph diffusion model can be generalized to other recommendation scenarios.

## D. Experiment Details

In this section, we provide details on the datasets, comparison methods, and hyper-parameter settings of comparison methods.

**D.1 Datasets**. We evaluate our experiments on the three real-world datasets from different domains: Yelp business dataset[15], Douban book dataset[55], and Flixster movie dataset[1]. All datasets contain user-item interaction information and social connections between users. The social connections in the Douban dataset are unilateral trust relationships, which means that when user A trusts user B, user B does not necessarily trust user A. Yelp and Flixter datasets have a friendship mechanism that is bilateral for both users.

**D.2 Comparison Methods**. We compare our GDSR with four group recommendation methods: (1) graph-based collaborative recommenders (i.e., LR-GCCF [2] and LightGCN [12]), (2) graph-based social recommenders (i.e., DiffNet++ [49] and GraphRec+ [7]), (3) denoising graph-based collaborative recommenders (i.e., RGCF [40], DDRM [58], and AdaGCL [17]), and (4) denoising graph-based social recommenders (i.e., GDMSR [34], DSL [43], GDSSL [20], and RecDiff [22]). We also note that DiffRec [45] and SGL [48] are denoising collaborative recommenders. Since DDRM outperforms DiffRec, and both AdaGCL and RGCF achieve better results than SGL, we do not include comparisons with DiffRec and SGL in our experiments. The descriptions for these baselines are listed as follows:

- LR-GCCF utilizes graph neural networks (GNNs) to model the user-item interaction bipartite graph. LR-GCCF first simplifies GNNs by removing non-linear activations, and then introduce a residual network structure for embedding combinations.
- LightGCN, similar to LR-GCCF, also models the user-item interaction graph using a simplified GNN structure. LightGCN removes activation functions and linear transformations, relying solely on neighborhood aggregation for layer-wise propagation.
- GraphRec+, is an upgraded version of GraphRec [6], which uses GNNs to model the interaction graph, social graph, and item relation graph constructed based on collaborative similarities.

---

[1]https://www.flixster.com/

- DiffNet++, an enhanced version of DiffNet [50], uses graph attention networks to model the user-item interaction graph and social graph separately.
- RGCF introduces a denoising method to alleviate noise issues from the user-item interaction graph, and then models user preferences for items based on the denoised interaction graph.
- DDRM applies diffusion models to denoise user and item embeddings. In this experiment, we use DDRM+SGL as the instantiation of DDRM due to its notable performance reported in their paper.
- AdaGCL is a graph collaborative filtering-based denoising method that incorporates data augmentation through graph denoising and generative model-based view generators.
- GDMSR designs a preference-guided graph denoising network to denoise the social graph, which generates recommendations based on the denoised social graph and user-item interaction information. We use DiffNet++ as the backend model for GDMSR.
- DSL alleviates noise issues in the social graph using a cross-view denoised self-supervision method and optimizes the recommender model using a multi-task learning strategy.
- GDSSL utilizes a diffusion model to generate a social subgraph structure. It then designs a contrastive learning task based on this subgraph to enhance the model representation learning.
- RecDiff applies a diffusion model to denoise user representations derived from the social graph, then combines the denoised representations with user-item interaction modeling.

***D.3 Hyper-parameter Settings***. Except for GraphRec+ [7] and GDSSL [20], all other baselines provided their code in their papers. For GraphRec+, we implement it based on the code in GraphRec [6]. For GDSSL, we implement this method ourselves. The settings of key hyper-parameters of these baselines are as follows:

- LR-GCCF: GNN layer size $L = 3$, embedding size $d = 64$.
- LightGCN: GNN layer size $L = 3$, embedding size $d = 64$.
- DiffNet++: GNN layer size $L = 2$, embedding size $d = 64$.
- GraphRec+: GNN layer size $L = 1$, MLP layer size $L' = 2$, embedding size $d = 64$, item similar neighbor size $k = 10$.
- RGCF: GNN layer size $L = 2$, embedding size $d = 64$, diversity loss coefficient $\lambda_1 = 0.000001$, temperature $\tau = 0.1$.
- DDRM: embedding size $d = 64$, diffusion step $T = 10$ (on Yelp and Flixster) or $T = 20$ (on Douban), noise scale $s = 0.001$, loss balance factor $\lambda = 0.2$, reweighted factor $\gamma = 0.9$.
- AdaGCL: GNN layer size $L = 2$, embedding size $d = 32$ (on Douban, Flixster) or $d = 64$ (on Yelp), SSL strength $\lambda_1 = 0.1$ (on Yelp, Douban) or $\lambda_1 = 0.01$ (on Flixster), temperature $\tau = 0.5$.
- GDMSR: GNN layer size $L = 2$, embedding size $d = 32$, co-optimization weight $\alpha = 0.5$, adaptive denoising factor $\gamma = 0.5$.
- DSL: GNN layer size $L = 2$, embedding size $d = 32$ (on Douban and Flixster) or $d = 64$ (on Yelp), SSL strength $\lambda_1 = 0.00001$ (on Yelp and Douban) or $\lambda_1 = 0.000001$ (on Flixster).
- GDSSL: GNN layer size $L = 2$, embedding size $d = 64$, diffusion step $T = 10$, SSL and social task strength $\lambda_1 = 0.01$, $\lambda_2 = 0.01$.
- RecDiff: GNN layer size $L = 2$, embedding size $d = 64$, timestep embedding dim $d' = 16$, diffusion step $T = 20$, noise scale $s = 0.1$.

# E. Contrastive Learning in Recommendation

In recent years, contrastive learning (CL) has emerged as a promising approach to enhance recommender systems [18, 56]. CL-based recommendation methods utilize additional supervision signals extracted from raw data, which can mitigate the data sparsity problem and improve model performance. The construction of contrastive views is crucial for CL-based recommenders. One line of current research [48, 53] uses data augmentation to create more views from the original data, while another line of studies [48, 52, 55] focuses on mining different views that exist in the data. Our GDSR aligns more closely with the first research line. Specifically, we first use diffusion models to enhance the original interaction bipartite graph and social graph, obtaining denoised graph structures. We then contrast the representations of user and item nodes in the graph before and after denoising. The node self-discrimination could provide auxiliary supervision signals for recommendation. We believe our diffusion-aware data augmentation paradigm will contribute to the advancement of CL-based recommender systems.

