# OpenReview forum: "Dual Graph Denoising Model for Social Recommendation"
_ACM.org/TheWebConf/2025/Conference — WWW 2025 Oral_

### Official Review · Reviewer_pgVc · 2024-11-07

**Novelty:** 4
**Technical Quality:** 5

**Review:**

This paper designs a graph diffusion social recommendation model. It introduces the diffusion model to perform graph denoising for the user-item graph and social graph, and then develops multiple conditions to guide the graph diffusion process.

Strengths:
(1) Problem statement is sufficiently motivated and the proposed methods have a degree of novelty.
(2) The paper is generally well written and organized. And the figures succinctly illustrate the model architectures and details.
(3) Experiments are detailed and complete. Experimental results show that the proposed model achieves significant performance improvements compared to baselines.
Weaknesses:
(1) Many of the model components have been reported in earlier research works like GDSSL [1] and RecDiff [2]. The technical novelty of this paper is not impressive.
(2) In the related work, the description of social recommendation and denoising recommendation is not comprehensive enough. This section lacks a systematic summary of the existing work.

[1] Jiuqiang Li and Hongjun Wang. 2024. Graph Diffusive Self-Supervised Learning for Social Recommendation. In SIGIR. 2442–2446.
[2] Zongwei Li, Lianghao Xia, and Chao Huang. 2024. RecDiff: Diffusion Model for
Social Recommendation. In CIKM 2024.

**Questions:**

1. If the initial vector of user neighborhoods C_0 contains 0 and 1, the values in the vector after diffusion are also 0 and 1?
2. Why select a set I_u^c containing k_c elements in Eq.14? How exactly do you choose these k_c elements?

**Reviewer Confidence:**

3: The reviewer is confident but not certain that the evaluation is correct

**Scope:**

3: The work is somewhat relevant to the Web and to the track, and is of narrow interest to a sub-community

---

### Official Review · Reviewer_J9Jj · 2024-11-12

**Novelty:** 3
**Technical Quality:** 4

**Review:**

In this paper, the authors propose a denoised social recommendation approach, named GDSR. In particular, GDSR designs two components: a diffusion-based graph denoising module and a recommendation module. The denoising module aims to filter potential noisy parts of the interaction and social graphs based on the elaborated dual diffusion models. Given the denoised graphs, the recommendation module learns user preference based on the denoised graphs. Experiments demonstrate the effectiveness of the proposed method.

Strengths:
- This paper is well-organized and easy to follow.
- The motivation is clear, and the intuition of introducing diffusion models to social recommendation denoising is reasonable.
- Experiments show the effectiveness of the proposed method.

Weaknesses:
- In my opinion, the main idea of introducing the diffusion process to recommendation denoising does not refresh. Existing works have established this paradigm on recommendation tasks, such as [1], [2]. So, what are the essential contributions of the proposed method? Maybe the authors can better highlight their contributions.
- The formulation of contrastive learning is not intuitive. Contrastive learning usually encourages consistency between two contrastive instances, so why use it to encourage consistency between the denoised and original graph embeddings? In fact, denoising meanwhile denotes the difference between two graphs.
- The compared baselines are not sufficient enough. For example, this paper combines social information and contrastive learning, so why not compare self-supervised social recommendations[3]? Besides, the latest social denoising recommendation methods are also not compared[4,5]. The authors should compare the most related baselines to validate the effectiveness of the proposed method.
- Besides performance improvements, how to show the denoising results? The authors should give a more detailed analysis of the denoised graphs.
- Section 4.7 is meanless. As we know, contrastive learning can improve the uniformity of node representations, this can not inspire us again.

[1] RecDiff: Diffusion Model for Social Recommendation, CIKM 2024.

[2] Denoising Diffusion Recommender Model, SIGIR 2024.

[3] Socially-aware self-supervised tri-training for recommendation, KDD 2021.

[4] Graph Bottlenecked Social Recommendation, KDD 2024.

[5] MADM: A Model-agnostic Denoising Module for Graph-based Social Recommendation, WSDM 2024.

**Questions:**

See weaknesses.

**Reviewer Confidence:**

3: The reviewer is confident but not certain that the evaluation is correct

**Scope:**

4: The work is relevant to the Web and to the track, and is of broad interest to the community

---

### Official Review · Reviewer_iKL2 · 2024-11-21

**Novelty:** 6
**Technical Quality:** 6

**Review:**

Summary:

The paper presents the Dual Graph Diffusion Model for Social Recommendation (GDSR), a novel approach that integrates graph denoising and user preference prediction to enhance the performance of social recommender systems. The GDSR model employs a dual diffusion mechanism to mitigate noise in user-item interaction and social graphs, followed by a dual graph learning structure for recommendation. The model also incorporates a diffusion-aware graph contrastive learning task to bolster the robustness of the recommendation module. The effectiveness of GDSR is demonstrated through extensive experiments on three public benchmark datasets, showing improvements over several state-of-the-art baselines.

### Pros:

1)	GDSR addresses a critical issue in social recommendation systems—noise in user interaction and social graphs—by introducing a dual diffusion model for denoising, which is a significant contribution to the field.
2)	The integration of a diffusion-aware graph contrastive learning task is innovative and enhances the model's representation quality and robustness.
3)	The model's performance is thoroughly evaluated, demonstrating its superiority over a range of strong baselines.

### Cons:

1)	In Appendix C.2, In Appendix C.2 Time Complexity, is the $b$ stands for the number of batches instead of batch size?
2)	while the paper compares GDSR with state-of-the-art methods in terms of performance, a direct comparison in terms of model and time complexity is missing. Such a comparison would provide a more comprehensive understanding of the trade-offs involved in using GDSR (not just analyze the GDSR’s own complexity)
3)	The introduction section's mention of "One school is based on data cleaning…" lacks clarity.
4)	The readability of the figures is compromised due to their small size, making it difficult to discern labels and text. Enlarging the figures would significantly improve the readability and accessibility of the visual content.

**Questions:**

Please address my concerns listed in the Cons.

**Reviewer Confidence:**

3: The reviewer is confident but not certain that the evaluation is correct

**Scope:**

4: The work is relevant to the Web and to the track, and is of broad interest to the community

---

### Official Review · Reviewer_1sxi · 2024-12-01

**Novelty:** 4
**Technical Quality:** 4

**Review:**

This paper presents GDSR (Graph Diffusion Social Recommender), a dual-model approach for social recommendation that addresses the challenge of noisy information in user-item interaction graphs and user-user social graphs. The proposed method consists of two main components: a denoising module that uses dual diffusion models (collaborative and social) to clean the graphs, and a recommendation module that employs a dual graph neural network structure with diffusion-aware graph contrastive learning to generate recommendations. The paper is easy to follow and presented well. The experiments provide valuable insights into the framework's performance and efficiency.

pros：

1. The paper introduces a dual-model architecture that separates denoising and recommendation tasks and introduce a dual GNN structure with a diffusion-aware graph contrastive learning paradigm to model user preferences.
2. The experimental results show good performance on three real-world datasets.

cons：

1. The motivation for this paper is not clearly presented in the INTRODUCTION, and the imperative of using diffusion models is not clearly explained. The approach in this paper appears to be a simple combination of previous social recommender model and diffusion models. Besides, diffusion models have been shown to improve the performance of social recommendations in previous studies[1,2,3], which indicates that the novelty of this paper is limited.

2. The paper appears to focus more on empirical results than theoretical justification, and the theoretical analysis of why the dual diffusion model works better is not described.

[1] Li, Z., et al. Recdiff: diffusion model for social recommendation. CIKM 2024.

[2] Wu, L., et al. Diffnet++: A neural influence and interest diffusion network for social recommendation. TKDE 2020.

[3] Wu, L., et al. A neural influence diffusion model for social recommendation.  SIGIR 2019.

**Questions:**

1. What is the rationale for using two diffusion models for noise reduction?  The motivation for using two diffusion models for noise reduction requires further explanation.
2. Is the approach in this paper an empirical result or based on theoretical justification?
3. According to Figure 5, the improvement in noise reduction in this paper is not significant compared to previous methods (although GDSR shows the smallest performance drop). Considering two diffusion models increases the model complexity, the contribution of the proposed method is not significant.

**Reviewer Confidence:**

4: The reviewer is certain that the evaluation is correct and very familiar with the relevant literature

**Scope:**

4: The work is relevant to the Web and to the track, and is of broad interest to the community

---

### Official Review · Reviewer_KXFx · 2024-12-02

**Novelty:** 5
**Technical Quality:** 5

**Review:**

From the perspective of the contributions of this paper, there are still some shortcomings. Although the author emphasizes the differences between the GDSR model and other related DM models in Section 5.2, the distinctions between them are rather limited. Furthermore, in terms of content, the introduction of the "Dual Graph" seems more like a straightforward application of DM models to the user-item interaction graph and the user-user social graph separately, and then combining them. The paper does not provide a clear explanation of why a Dual Graph is necessary instead of using a standalone user-item interaction graph or a user-user social graph.

Regarding the content of the paper, the structure is relatively clear. One commendable aspect is that the figures representing the model and the ablation experiments are visually appealing and clear. The experimental work is also thorough. However, there is a small issue worth mentioning: the tables reflecting the experimental results do not include any annotations for the second-best results, and the 13th and 14th references in the bibliography refer to the same paper. The author should pay attention to these minor details.

**Questions:**

1: Why use "Dual Graph"?
2: In Section 5.2, the authors mentioned that the difference between the proposed model and RecDiff, GDSSL is that "they do not incorporate condition guidance relevant to the social recommendation task in their DM design." Are there any other related baselines for the social recommendation task that could further demonstrate the superiority of the proposed model?

**Reviewer Confidence:**

4: The reviewer is certain that the evaluation is correct and very familiar with the relevant literature

**Scope:**

4: The work is relevant to the Web and to the track, and is of broad interest to the community